# Effects of mycotoxin-producing fungi on the fitness and gut bacterial community of the soil springtail *Folsomia candida*

Yang Xu,[1] Lingxiao Tang,[1] Zhen Xie,[2] Xingwei Duan,[1] Kaisha Wang,[1] Jialin Zhu,[1] Yangyang Huang,[1] Kailang Yang,[1] Lei Xu,[2] Hong He[1]

**ABSTRACT** Mycotoxin-producing fungi are widespread and their adverse effects on mammals have been investigated; however, their impacts on soil invertebrates are not fully understood. *Folsomia candida* is a model soil arthropod that represents an important part of the soil invertebrate community. This study investigated the consequences of *F. candida* grazing on mycotoxin-producing fungi *Fusarium verticillioides*, *F. graminearum*, *Aspergillus ochraceus*, and *A. nidulans*. Consuming mycotoxin-producing fungi affected the body size and reproductive ability of *F. candida*, and altered the gut bacterial composition, with decreased Proteobacteria and increased Actinobacteria (*Microbacterium*) abundances. Notably, the abundance of foodborne fungi can be detected. Furthermore, certain bacteria isolated from *F. candida*'s gut inhibited the growth of corresponding mycotoxin-producing fungi. The gut bacteria that inhibited mycotoxin-producing fungi growth in *Aspergillus* groups were also associated with poor fitness parameters and larger disruption in gut microbiota. Importantly, switching back to yeast diets reversed both the fitness parameters and gut bacterial composition. Together, our study demonstrated that grazing of mycotoxin-producing fungi by *F. candida* resulted in reduced physiological parameters and disturbed the gut bacterial community, and those changes can be restored by switching back to yeast diets, which indicates a strong resilience of springtails to mycotoxin-producing fungi.

**IMPORTANCE** Mycotoxin-producing fungi are widespread in nature and raise concerns for human and livestock health. Although they share the same ecosystem, interactions between mycotoxin-producing fungi and soil arthropods are not well understood. In this study, we report an unexpected finding that the soil arthropod *Folsomia candida* is rather tolerant to these mycotoxin-producing fungi. *F. candida* can survive solely on mycotoxin-producing fungi as a food source with reduced physiological parameters. Moreover, the gut microbial community is disturbed by mycotoxin-producing fungi, and some of the bacteria isolated from *F. candida*'s gut can inhibit the growth of corresponding fungi. Notably, the altered physiological parameters and gut microbiota are restored when a normal diet is reintroduced, suggesting *F. candida*'s resilience to mycotoxin-producing fungi. These findings clarify the impact of toxin-producing diets on *F. candida*, shedding light on how organisms can build resilience to environmental stimuli.

**KEYWORDS** *Folsomia candida*, mycotoxin-producing fungi, fitness parameter, gut bacteria, resilience

Fungi like *Fusarium* sp. and *Aspergillus* sp. produce various fungal toxins including fumonisins, deoxynivalenol, zearalenol, sterigmatocystin, and aflatoxins, induced by environmental stress, such as temperature, water activity (1), reactive oxygen species (2), light, carbon source (3), and animal predation (4). These fungal toxins are toxic fungal secondary metabolites that act as defense compounds with carcinogenic, mutagenic, teratogenic, and hepatotoxic effects (5–7). Exposure to fungal toxins *via* consumption

Address correspondence to Hong He, hehong@nwsuaf.edu.cn, or Lei Xu, xulei@nwafu.edu.cn.

Yang Xu, Lingxiao Tang, and Zhen Xie contributed equally to this article. Author order was determined by degree of contribution to the manuscript.

The authors declare no conflict of interest.

See the funding table on p. 14.

of mold-contaminated diets can lead to severe diseases in mammals (8–10). However, many invertebrates exhibit a high tolerance for mycotoxin-producing fungi and their metabolites (11, 12). For example, *Fusarium* species are barely toxic to invertebrates, as some species are the preferred diet for *Folsomia candida, Heteromurus nitidus, Protaphorura armata,* and *Tenebrio molitor*, and increase the springtails populations (13–15). Some *Aspergillus* species exhibit relatively lower toxicity to springtails, as they cause little mortality, although animal growth and reproduction are impaired (4, 11). A wide range of invertebrates has been proven to show high tolerance to aflatoxin B1, zearalenone, deoxynivalenol, and ochratoxin A (16–18). Considering the potentially toxic effects of these secondary metabolites on mammals, it is intriguing to further investigate the mechanism behind the coexistence of invertebrates with mycotoxin-producing fungi (19).

Gut bacteria have a crucial role in maintaining the health of their host (20). Animal gut microbes have a beneficial effect on their nutritional assimilation, detoxification, and immune systems (21–25). The gut microbiota is highly adaptable to different developmental stages and can be influenced by elements including diet, host genetics, and ecological factors (26, 27). Notably, diets have been recognized as the most important factor that shapes the architecture and functionality of the gut microbiota (28, 29). Furthermore, exposure to diverse diet treatments affects not only the gut microbiota but also changes urine metabolic profiles, stool bacterial functional genes abundance, and functional cell proportion (30, 31). Therefore, altered gut microbiota and specific genera may have significant roles in maintaining gut microbiota function against stressful factors.

*F. candida* is a commonly used model organism in ecotoxicity studies to explore the interaction between invertebrates and the environment (32, 33). The gut microbiota of *F. candida* is a repository of microbial biodiversity and shapes the interactions between *F. candida* and the soil environment (34–36). The composition of *F. candida* gut microbiota can be affected by diverse stimuli like antibiotics (37), Ag Nanoparticles (38), and liming (36). It is believed that diet is also the key factor regulating the gut microbial composition of *F. candida* (39). In the soil ecosystem, *F. candida* consumes diverse food such as plant litter, bacteria, and fungi (11, 32, 40–42). Among these, fungi are considered a preferred food of *F. candida* in soil ecosystems, springtails have been even evaluated as biological control agents against mycotoxin-producing fungi such as *Fusarium* spp (14, 15, 43, 44). In addition, feeding *F. candida* with another mycotoxin-producing fungus *Aspergillus* harmed the animal's growth and reproduction rate, but mortality was not significantly impacted (11). It is not clear whether the fitness parameters and microbial community of *F. candida*'s gut are affected when they consume various mycotoxin-producing fungi. Furthermore, the underlying mechanisms of gut microbiota in facilitating *F. candida* to minimize the toxicity of fungi secondary metabolites remain enigmatic.

In this study, we investigated the impact of four different mycotoxin-producing fungi *F. verticillioides*, *F. graminearum*, *Aspergillus ochraceus,* and *A. nidulans* on the fitness parameters and gut microbiota of *F. candida*. To mimic natural conditions, we exposed springtails to mycotoxin-producing fungi cultured on rice medium. By presenting mycotoxin-producing fungi as the sole diet for *F. candida*, we found that grazing those fungi altered its fitness parameters and gut microbial communities. Moreover, we observed that these changes could be recovered once the diet was switched back to a regular yeast diet. More importantly, we isolated culturable microorganisms that could hinder the growth of corresponding fungi, suggesting a counteracting mechanism of gut bacterial communities against mycotoxin-producing fungi. In conclusion, our findings demonstrated that *F. candida* is resistant to mycotoxin-producing fungi, and its gut microbiota underwent significant changes after grazing on such fungi.

## MATERIALS AND METHODS

### Fungi strain and collembolan

The collembolan *F. candida* was kindly provided by Dr. Liang Chang (Northeast Institute of Geography and Agroecology, Chinese Academy of Sciences) and was kept as previously described (42). *F. candida* was kept on Petri dishes filled with a layer of gypsum plaster with charcoal (9: 1) with proper humidity. They were routinely fed with yeast and kept at room temperature (15°C–25°C). All experimental *F. candida* individuals had been size-synchronized and pre-starved for 2 days in Petri dishes before diet manipulation to increase their appetite. Fungi strains *F. graminearum*, *F. verticillioides*, *A. ochraceus*, and *A. nidulans* were kindly provided by Prof. Dr. Karlovsky (Goettingen University, Germany). Baker's yeast (*Saccharomyces cerevisiae*) was purchased as a commercial product (TH-AADY, diploid, thermal-tolerant alcohol active dry yeast, Angel Yeast Co., Ltd., Yichang, China, http://www.angelyeast.com). Mycotoxin-producing fungi diets were made by culturing each fungus on a rice medium incubated at 25°C under dark conditions, and the rice medium was made by the addition of 0.5 g rice powder and 1.65 mL tap water to glass Petri dishes (6 cm in diameter) and autoclaved (42).

### Exposure of *F. candida* to different dietary treatments

In the feeding experiment, five diets were served as the only diet for the first generation of *F. candida* (F1 *F. candida*): (i) *F. graminearum,* (ii) *F. verticillioides,* (iii) *A. ochraceus,* (iv) *A. nidulans,* (v) and yeast (positive control). Collembola starved served as the negative control. All treatments were performed in a 90 mm × 15 mm Petri dish with a layer of plaster to keep proper humidity and each treatment contained three replicates. A total of 18 experiment arenas were built. Each assay was introduced with a total of 30 specimens of *F. candida* of similar size (initial body length at 1.53 ± SE 0.15 mm and head width at 0.30 ± SE 0.03 mm). Each diet was served by allocating a piece of 7-day-old mycotoxin-producing fungi or yeast to the center of the plaster medium as the only food source for springtails, and no food was provided for the hunger group. The experiment was performed at room temperature in a constantly dark condition and supplied with enough food and tap water once a week. After 4 weeks, the growth parameter of first-generation and second-generation *F. candida* individuals (F1 and F2) was recorded and 25–30 F1 *F. candida* guts were collected and used for DNA extraction.

The second feeding experiment begins with transferring 30 ± 2 F2 *F. candida* grown on mycotoxin-producing fungi for four weeks to a new Petri dish containing plaster medium with proper humidity. Three replicates were prepared from each condition as well. They were all fed with yeast for 6 weeks and the diet was replaced every week. The fitness parameter of *F. candida* (F2 and F3) was recorded and 25–30 F2 *F. candida* guts were collected.

### Analysis of fitness parameter of *F. candida*

To calculate the survival rate of *F. candida*, the dead body of the first introduced 30 *F. candida* in each feeding experiment was removed and counted weekly. At the end point of each feeding experiment, collembola was measured by taking digital pictures using a stereoscope with a piece of coordinate paper manually added (Leica, Germany). After that, their body length and width were measured using Image J software (https://imagej.net/) (45). The number of living adults and juveniles was counted by taking photographs of the animals floating on the water surface with several drops of blue ink added, and the number of springtails was counted on the photos.

### *F. candida* gut microorganism analysis

In both feeding experiments, after exposure to a different diet, 25–30 adults were collected from each experiment assay for gut DNA extraction. The collembolans were first washed with 75% ethanol solution for 10 s and then rinsed twice in sterile water.

The guts dissected with sterile forceps in a sterile environment were placed in a 1.5 mL centrifuge tube. 25–30 digestive tracts were pooled and stored at −20℃ until DNA extraction. The genomic DNA was extracted using a Magpure stool DNA KF kit (Magen, China), and the nucleic acid quality was evaluated using 1% agarose gel. Finally, the extracted gut DNA was stored at −20℃ until analysis.

V3-V4 of bacterial 16S rRNA gene was amplified with degenerate PCR primers, 341F/806R, while the ITS2 of the Internal Transcribed Spacer (ITS) region was amplified with degenerate PCR primers ITS 3 and ITS 4 (Table S1). PCR cycling conditions were as follows: 94℃ for 3 min, 30 cycles of 94℃ for 30 s, 56℃ (for bacteria) or 58℃ (for fungi) for 45 s, 72℃ for 45 s, and final extension for 10 min at 72℃. The PCR products were purified with AmpureXP beads and eluted in the Elution buffer. Libraries were qualified by the Agilent 2100 bioanalyzer (Agilent, USA). The validated libraries were used for sequencing on the Illumina MiSeq platform (BGI, Shenzhen, China) following the standard pipelines of Illumina, and generating $2 \times 300$ bp paired-end reads.

Raw reads were filtered to remove adaptors and low-quality and ambiguous bases before paired-end reads were added to tags using the Fast Length Adjustment of Short reads program (FLASH, v1.2.11) (46). The tags were clustered into OTUs with a cutoff value of 97% using UPARSE software (v7 0.0.1090) (47) and chimera sequences were compared with the UNITE (v20140703) using UCHIME (v4.2.40) (48). Then, OTU representative sequences were taxonomically classified using Ribosomal Database Project (RDP) Classifier v.2.2 with a minimum confidence threshold of 0.6 and trained on the UNITE (V6 20140910) by QIIME v1.8.0 (49). The USEARCH_global was used to compare all Tags back to OTU to get the OTU abundance statistics table of each sample (50).

## Isolation of cultivable microbes

The digestive tract of *F. candida* fed on different diets and hunger in the first feeding experiment was dissected as above. Guts from 25 to 30 adults from each treatment group were pooled and 100 µL sterile water was added and homogenized with a sterile pellet pestle. Then they were diluted to proper concentration and spread 100 µL diluents on plates of potato dextrose agar, Luria-Bertani Agar, and Gauze's Synthetic Agar medium (Aoboxing Biology Technology, Beijing). Plates were incubated in a growth chamber at 25℃ under dark conditions. After 2–3 days of culturing, the different microbial colonies were chosen according to their distinct morphology, including shape, elevation, surface, size, opacity, and pigmentation. The screened microbial colonies were further identified by performing DNA extraction and PCR amplification using bacterial 16S rRNA primers (Ba27F and Ba1492R) (Table S1). At least two representatives of each kind of cultured microorganism colony were sequenced.

## Inhibition of fungi by isolated gut microorganisms

In total, the mycotoxin-producing fungi inhibition ability of 19 bacteria was tested. Nine replicates were prepared for each isolated bacteria strain to investigate the bacterial role in inhibiting their corresponding mycotoxin-producing fungi growth. In the beginning, mycotoxin-producing fungi spore suspension (1,000 spores/µL) was inoculated and grown in 30 mL of PDB medium with 180 rpm at 25℃ for 1–2 days, and three replicates were harvested as initial control (2d ck). After that, 1 mL of fresh LB medium containing isolated bacteria (OD = 0.9) was added to another three replicates, while the same volume of pure LB medium added to the rest three replicates served as the control (5d ck). All treatments were incubated at 180 rpm at 25℃ under dark conditions, and the samples were harvested after 3 days. All harvested samples were centrifuged at 5,000 rpm for 10 min, and the sediment was collected and freeze-dried. The biomass of fungi or dual cultures was determined by the weight of the freeze-dried material. DNA of dual culture was extracted from 10 mg freeze-dried material and kept at −20℃ until use (51).

## qPCR assay

DNA of the gut bacterial/fungal abundance and dual culture were employed to determine their absolute abundance using quantitative PCR. All qPCRs were carried out on the CFX Connect Real-Time PCR Detection System (Bio-Rad, USA) with primers listed in Table S1 (341F and 806R for bacteria). Standards for target genes were prepared according to the previous study by amplification of the genomic DNA of each mycotoxin-producing fungi or *E. coil* (52, 53). Each reaction was performed in duplicate on the same plate in the total volume of 10 µL (0.2 µM of each forward and reverse primer; SYBR Green Premix Pro Taq HS qPCR kit, Accurate Biotechnology) with 1 µL of DNA.

## Statistical analysis

Experimental data analyzed for significance were performed using GraphPad Prism 7 (GraphPad Software, San Diego California USA). Statistic methods used include unpaired, two-tailed Student's t test, one-way or two-way ANOVA test, or Mann Whitney test, and are indicated for all figures. Error bars represent ± SEM. $*P < 0.05$; $**P < 0.01$; $***P < 0.001$, $****P < 0.0001$. The linear discriminant analysis effect size (LEfse) method was utilized to compare significant differences in taxa between groups.

## RESULTS

### Mycotoxin-producing fungi inhibit the growth of springtails

The diet plays a crucial role in regulating the physiological state of *F. candida* (11, 54). The food preference assay showed *F. candida*, as a fungivorous species, prefers *F. verticillioides*, *F. graminearum*, *A. ochraceus,* and *A. nidulans* as their diet compared to the rice medium that supports the consumption of mycotoxin-producing fungi than substrate medium by springtails (Fig. S1). We found that *F. candida* could graze mycotoxin-producing fungi (Fig. 1a; Table S3). Analysis of the physiological parameters of springtails revealed an impact of mycotoxin-producing fungi on animal performance. Specifically, the feeding of *A. nidulans* and *A. ochraceus* led to significantly lower body length and width of first-generation (F1) adults (Fig. 1b). However, no significant differences in the mean body size of F1 adults between the *Fusarium* feeding groups and the yeast control group were detected (Fig. 1b). Interestingly, a profound reduction in body width and length in the second generation (F2) that fed on all mycotoxin-producing fungi-feeding groups (Fig. 1a through c). To determine whether feeding on different fungi affects the survival of *F. candida*, we calculated the number of dead F1 adults in the 4-week feeding test in different groups. Unexpectedly, the overall survival rate of *F. candida* was not influenced by these mycotoxin-producing fungi and was even higher in the *A. nidulans* group (Fig. 1d), suggesting that *F. candida* exhibits tolerance to mycotoxin-producing fungi. Next, we observed that mycotoxin-producing fungi affect the reproduction of *F. candida*. Specifically, *F. candida* had a significantly higher reproduction ability when fed with *Fusarium,* and the juvenile numbers were significantly increased in those groups (Fig. 1e). In addition, a similar number of juveniles were observed in the *Aspergillus*-fed groups. Together, we showed that *F. candida* exhibits a high tolerance to mycotoxin-producing fungi but its body size, especially juvenile body size was reduced.

### Foodborne fungi abundance of F1 *F. candida* grazing on mycotoxin-producing fungi

To explore whether *F. candida* truly consumes mycotoxin-producing fungi or not, the relative abundance of OTU belonging to foodborne fungi genera in the digestive tract was determined. Digestive tracts from 25 to 30 adults were collected and gut DNA was extracted. The ITS2 regions were acquired with three replicates of *F. candida* gut with different treatments for 4 weeks. We measured the relative abundance of OTU belonging to *Saccharomyces* in the yeast group, *Fusarium* in *F. verticillioides* and *F. graminearum* groups, and *Aspergillus* abundance in *A. nidulans* and *A. ochraceus* groups (Fig. 2a). The

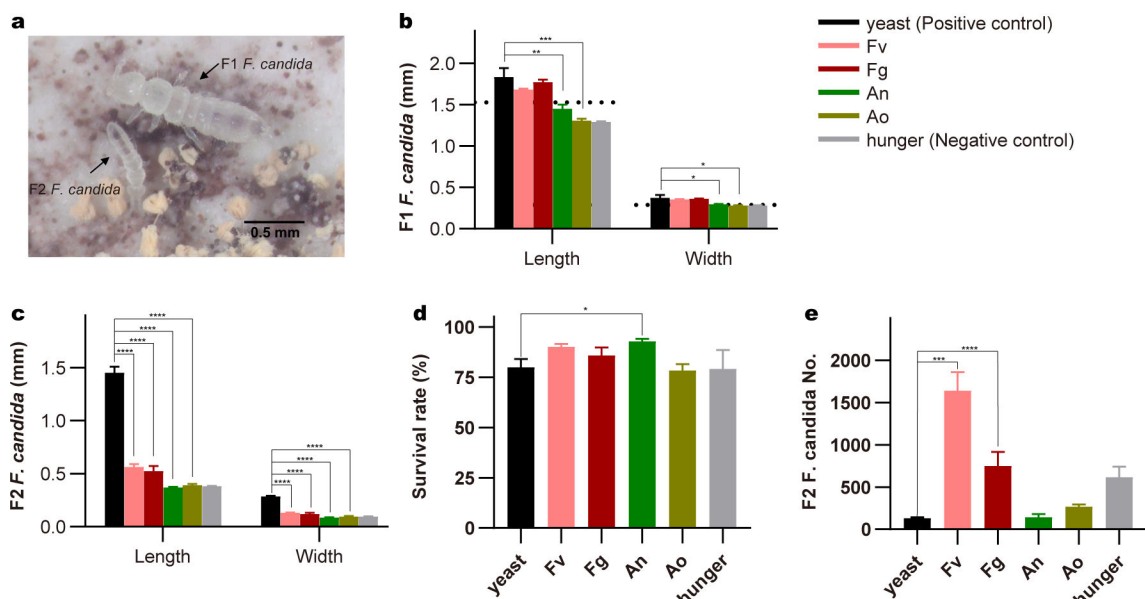

**FIG 1** Physical parameters of *F. candida* feed on various mycotoxin-producing fungi. (a) Representative image of F1 (first generation) and F2 (second generation) *F. candida* grazing on *A. ochraceus*. Body length and width of F1 (b) and F2 (c) *F. candida* feed on yeast, *F. verticillioides* (Fv), *F. graminearum* (Fg), *A. nidulans* (An), *A. ochraceus* (Ao), or Hunger. Dotted lines indicate the initial size (length and width) of F1 *F. candida*. The survival rate of the F1 *F. candida* (d) and the number of F2 *F. candida* (e) fed on different diets. Data are shown as mean ± SEM (*n* = 3). *P* values were calculated using one-way ANOVA followed by Dunnett's multiple comparisons test. *$P < 0.05$; **$P < 0.01$; ***$P < 0.001$.

corresponding diet fungi genera can be detected in nearly all the digestive tract of *F. candida*, except for one replicate of the *F. veriticillioides* group. The most abundant foodborne fungi detected were in *the A. ochraceus* group, with *Aspergillus* genera relative abundance reaching 69.22%, followed by *A. nidulans* group with *Aspergillus* genera abundance present as 10.09%. The abundance of *Fusarium* genera in *F. verticillioides* and *F. graminearum* were much lower, at 4.87% and 0.81%. Yeast abundance was the lowest foodborne fungi can be detected, at 0.18%. The sequencing results demonstrated that the relative abundance of foodborne fungi in *F. candida* was significantly different from each other, these results may suggest *F. candida* has different digestion capacities for yeast and various mycotoxin-producing fungi.

## Bacterial communities of F1 *F. candida* fed on mycotoxin-producing fungi

To explore whether the feeding of mycotoxin-producing fungi could alter the gut bacterial community of *F. candida*, the bacterial diversity and community composition in the digestive tract were analyzed. The impact of exposure to mycotoxin-producing fungi on gut bacterial communities was assessed using beta diversity analysis based on weighed_unifrac and PCA plots based on the Bray-Curtis distance. The results revealed significant separation among the gut bacterial communities of F1 *F. candida* fed on different mycotoxin-producing fungi, yeast, and starvation (Fig. S2). The sequencing results demonstrated that the diversity and community composition of gut bacteria in *F. candida* were significantly altered by grazing mycotoxin-producing fungi.

Consistent with the previous finding that showed Proteobacteria were the dominant phylum accounting for >50% of the bacterial abundance in *F. candida* gut (35, 55), we also noticed that Proteobacteria represented the most abundant phylum considering all groups (average across all groups, 66.01%), followed by Actinobacteria (19.30%) and Bacteroidetes (6.83%) (Fig. 2b). In the yeast group, the most abundant phylum was Proteobacteria, followed by Bacteroidetes and Actinobacteria. In the hunger group, the most abundant taxa were Proteobacteria, the proportion was lower compared with the yeast group. The following abundant taxa were Actinobacteria and Firmicutes. In *F.*

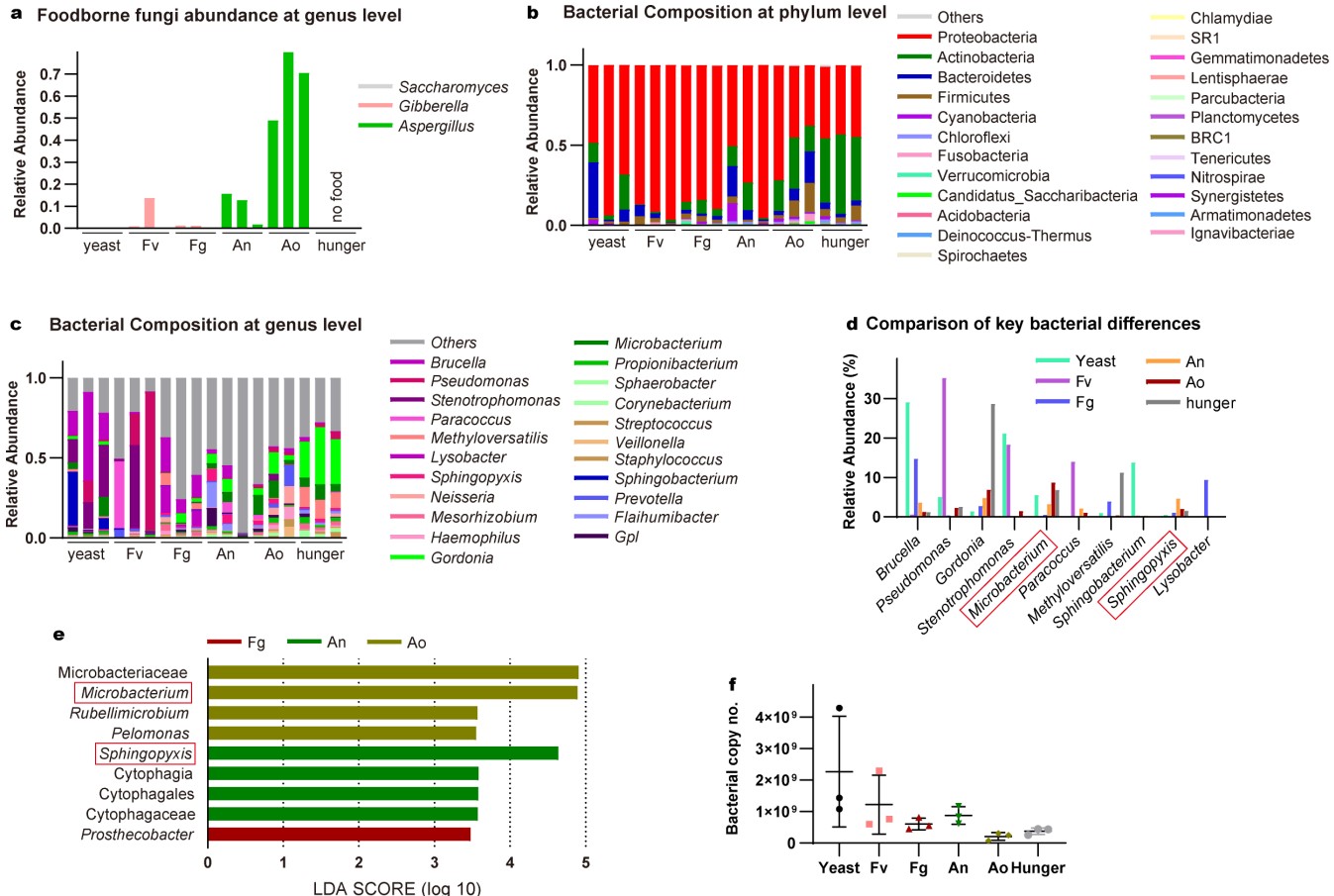

**FIG 2** Gut microbiol community of F1 *F. candida* feed on mycotoxin-producing fungi. (a) The relative abundance of foodborne fungi at the genus level. The gut bacterial communities of F1 *F. candida* feed on yeast, mycotoxin-producing fungi, and hungered are classified at the phylum level (**b**) and genus level (**c**). (d) Relative abundance of the top 10 key bacteria in F1 *F. candida* gut microbial community at the genus level. (e) LEfSe analysis of microbial abundance between the yeast and mycotoxin-producing fungi feeding groups. (f) Gut bacterial content of 25–30 *F. candida* in different treatments (*P* values calculated using one-way ANOVA followed by Dunnett's multiple comparisons test).

*verticillioides* and *F. graminearum* groups, Proteobacteria was the most dominant phylum, with Actinobacteria as the second abundant phylum. In the *F. verticillioides* group, Bacteroidetes was the third abundant phylum but in the *F. graminearum* group was Firmicutes. In the *A. nidulans* group, the most abundant phylum was Proteobacteria, followed by Actinobacteria and Bacteroidetes. In the *A. ochraceus* group, the proportion of Proteobacteria was decreased, while the proportion of Actinobacteria was increased compared to other groups (Table S4). We also noticed Cyanobacteria, Chloroflexi, and Fusobacteria were detected in minor proportions.

We further analyzed the bacterial community at the genus level (Fig. S3). The genus composition exhibited clear differences among groups, with *Gordonia* (average across all groups, 10.58%), *Brucella* (8.39%), *Pseudomonas* (7.62%), *Stenotrophomonas* (6.82%), and *Microbacterium* (4.29%) representing the core gut microbiota of *F. candida* (Fig. 2c). In the yeast group, *Brucella* and *Stenotrophomonas* were detected at high abundances, followed by *Sphingobactertium*. In the hunger group, *Gordonia* was dominant, followed by *Methyloversatilis* and *Microbacterium*. The *F. verticillioides* group had the most abundant genus of *Pseudomonas*, followed by *Gordonia* and *Stenotrophomonas*. In the *F. graminearum* group, *Brucella* was the most abundant genus followed by *Lysobacter* and *Methyloversatilis*. In the *A. nidulans* group, the dominant groups were *Flavihumibacter*, *Gordonia,* and *Sphingopyxis*. In the *A. ochraceus* group, *Microbacterium* was predominant, followed by *Gordonia* and *Prevotella* (Table S5). In addition, the absolute abundance

of bacterial DNA amount was reduced when those mycotoxin-producing fungi were provided as the only food source, although the changes are not significant (Fig. 2f). Together, our findings provide evidence that mycotoxin-producing fungi significantly altered the gut bacterial community composition of *F. candida*.

LEfse method revealed that the genera *Microbacterium*, *Rubellimicrobium*, and *Pelomonas* enriching in Ao group; the genus *Sphingopyxis,* and the class Cytophagia as well as their derivative are significantly higher in the An group; *Prosthecobacter* abundance are significantly higher in Fg group (Fig. 2e). Notably, *Microbacterium* and *Spgingopyxis* are the top 10 key bacteria in the collembolan gut microbial community in different treatments. In terms of numbers as well as abundance of differed bacteria (Fig. 2d), we can see that *Aspergillus* groups are the most disrupted groups, while *Fusarium* groups are less disrupted.

## Culturable microbiota of the gut of F1 *F. candida* feed on mycotoxin-producing fungi

We next attempted to isolate culturable bacteria to address the role of those bacteria in the resistant mechanisms of *F. candida* against mycotoxin-producing fungi. In total, 33 cultured bacterial strains isolated from the gut of *F. candida* feed on different diets (Fig. 3a and b). All of the bacteria belong to Actinobacteria, Firmicutes, and Proteobacteria, which contains 12 bacterial genera: *Microbacterium*, *Galactobacter*, *Brachybacterium*, *Pimelobacter*, *Streptomyces*, *Mammaliicoccus*, *Staphylococcus*, *Bacillus*, *Stenotrophomonas*, *Acinetobacter*, *Brucella,* and *Paracoccus*. The distribution of the microorganism species was also different in mycotoxin-producing fungi groups. *Microbacterium* was found in all the treatment groups, except for the yeast group. *Galactobacter* was isolated from yeast, *Aspergillus,* and hunger groups. In addition, the dominant bacterial genera in each group were different, with *Bacillus* dominant in the yeast group, *Microbacterium* dominant in mycotoxin-producing fungi feeding groups, and *Streptomyces* dominant in the Hunger group. Overall, the most abundant and versatile gut microbes were detected in the yeast-feeding and hunger groups, while mycotoxin-producing fungi-feeding groups showed lower gut microbe diversity. These results further proved that mycotoxin-producing fungi affect the abundance and diversity of *F. candida* gut bacteria.

## Bacteria isolated from F1 *F. candida* gut inhibit *Aspergillus* growth

To assess whether the gut microbes isolated from F1 *F. candida* are capable of inhibiting the growth of mycotoxin-producing fungi, we co-cultured these bacteria with their corresponding diet fungi species. The biomass of fungi was monitored by determining the total DNA level of mycotoxin-producing fungi. For each sample, 10 mg of freeze-dried samples was weighted for DNA extraction and the DNA copy number was quantified by qPCR. Compared to the control (individual mycotoxin-producing fungi culture), the co-culture with *Microbacterium*, *Streptomycetaceae*, *Acinetobacter*, *Brucella*,

**a** Composition of culturable bacteria at Phylum level

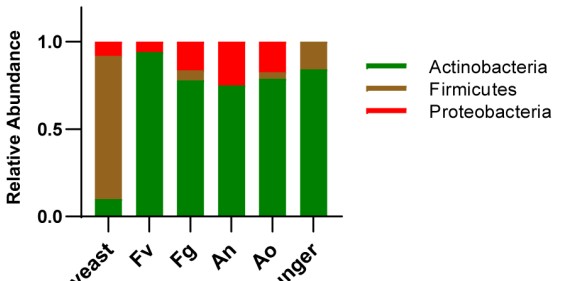

**b** Composition of culturable bacteria at genus level

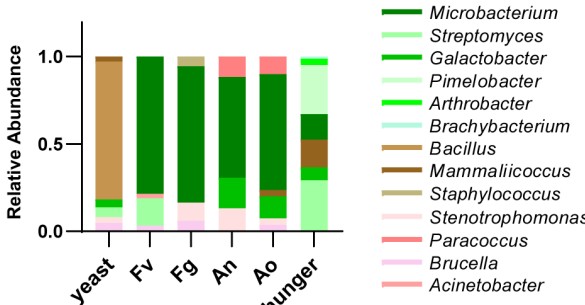

**FIG 3** Culturable bacteria isolated from F1 *F. candida* feed on different diets. Culturable bacteria isolated from F1 *F. candida* feed on yeast, mycotoxin-producing fungi, and hungered are classified at the phylum level (a) or genus level (b).

and *Staphylococcus* did not show any inhibitory effect on the biomass of *F. verticillioides* and *F. graminearum* as the mycotoxin-producing fungi content is higher or similar compared with 5d ck (Fig. 4a and b). By contrast, the total DNA level of *A. nidulans* was reduced when it was co-cultured with *Microbacterium*, *Paracoccus*, *Galactobacter*, and *Stenotrophomonas* (Fig. 4c). Notably, *Galactobacter* and *Microbacterium* showed a stronger inhibitory effect than other bacteria compared to 5d CK. Furthermore, we found that *Brucella* and *Stenotrophomonas* could also inhibit the growth of *A. ochraceus*, while *Galactobacter*, *Microbacterium*, and *Mammaliicoccus* showed no effect or enhanced mycotoxin-producing fungi growth. Interestingly, *Arthrobacter* and *Paracoccus* even stimulated the growth of *A. ochraceus* (Fig. 4d). Together, our results indicated that certain gut bacteria of *F. candida* have the potential to inhibit the growth of *A. nidulans* and *A. ochraceus*.

## The second generation of *F. candida* regained growth and reproduction potential

To investigate whether the negative effects of mycotoxin-producing fungi on the growth and reproduction of F2 *F. candida* were reversible, F2 *F. candida* fed with a mycotoxin-producing diet was reintroduced with normal diet yeast, and their fitness parameters were measured. After the restoration of yeast diets for 6 weeks, the body length and width of F2 *F. candida* originating from mycotoxin-producing fungi groups were the same or bigger as that of groups fed on yeast (Fig. 5a). Furthermore, the overall survival rate of F2 *F. candida* was also comparable to the yeast group (Fig. 5b), while the offspring number was lower than the yeast group, especially in *F. verticillioides* and *A. nidulans* groups (Fig. 5c). Together, these results suggest that growth and reproduction reduction of *F. candida* caused by mycotoxin-producing fungi can be recovered at a large extent when normal diets are provided.

## Foodborne fungi abundance of F2 *F. candida* after feeding with normal diets

To explore whether *F. candida* could recover from mycotoxin-producing fungi or not, the relative abundance of mycotoxin-producing fungi abundance in the digestive tract was determined in *F. candida* that was fed with a normal yeast diet after being exposed to mycotoxin-producing diets was also analyzed (Fig. 6a). The result showed that OTU belonging to genus *Fusarium* and *Aspergillus* in corresponding groups still can be detected, but with times lower abundance compared to the first feeding experiment, 0.24% in *F. verticillioides* groups, 0.19% in *F. graminearum* group, 0.32% in *A. nidulans,* and 0.24% in *A. ochraceus*. Meanwhile, the abundance of foodborne yeast diet in every sample was also examined as well. The results showed that yeast diet can be detected in every sample, but similar to the first feeding experiment, yeast abundance is low ranging

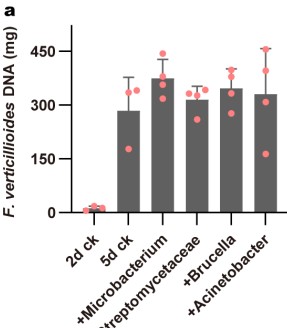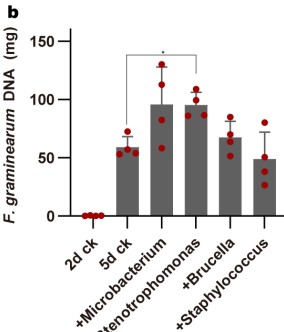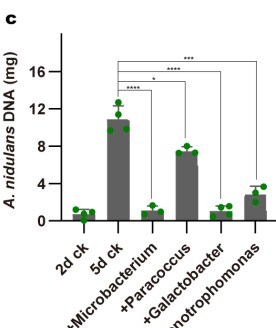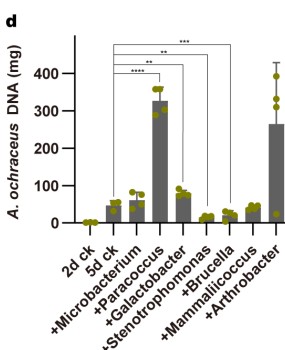

**FIG 4** Mycotoxin-producing fungi content in their dual culture with corresponding culturable bacteria. The DNA amount of *F. verticillioides* (a), *F. graminearum* (b), *A. nidulans* (c), and *A. ochraceus* (d) that co-cultured with different bacteria isolated from F1 *F. candida* gut feeding on corresponding mycotoxin-producing fungi. Data are shown as mean ± SEM (*n* = 3). *P* values calculated using one-way ANOVA followed by Dunnett's multiple comparisons test. *$P < 0.05$; **$P < 0.01$; ***$P < 0.001$; ****$P < 0.0001$.

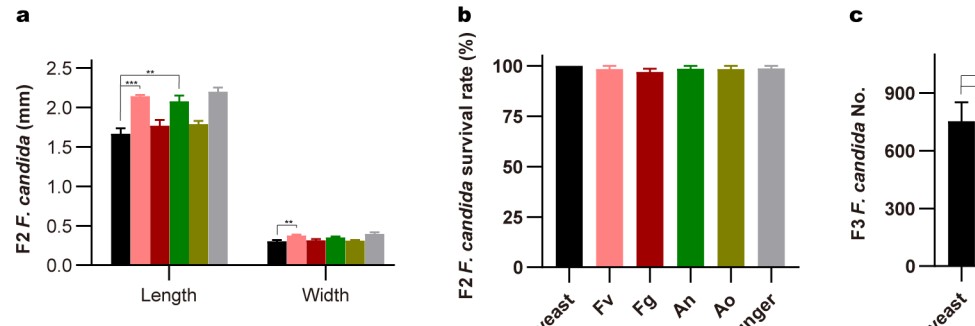

**FIG 5** Physical parameter of F2 *F. candida* after each treatment was reintroduced with the yeast diet. Body length and width of F2/F3 (third generation) *F. candida* (a), the survival rate of the F2 *F. candida* (b) and the number of F3 *F. candida* (c) fed on yeast diets for 6 weeks that originated from the F1 *F. candida* grazing yeast, *F. verticillioides*, *F. graminearum*, *A. ochraceus*, *A. nidulans* diets, or hunger. Data are shown as mean ± SEM (*n* = 3). *P* values calculated using one-way ANOVA followed by Dunnett's multiple comparisons test. **P* < 0.05; ***P* < 0.01; ****P* < 0.001.

from 0.08% to 2.25%. Together, these results demonstrated that the abundance of *F. verticillioides*, *F. graminearum,* and *A. nidulans* was 4–31 times (19, 4, or 31 times, respectively) lower in F2 *F. candida* gut that has been reintroduced with a yeast diet, while the abundance of *A. ochraceus* is significantly reduced in F2 *F. candida* (Table S6).

## Gut bacterial communities of F2 *F. candida* after feeding with normal diets

Next, the gut microbiome of F2 *F. candida* fed with a normal yeast diet after fungal toxins exposure was analyzed (Fig. 6). The results showed that the main phyla in the gut microbiome of F2 *F. candida* were Proteobacteria, Firmicutes, Actinobacteria, and Bacteroidetes (Fig. 6b). Interestingly, no significant differences in bacterial community composition were detected among those groups in F2 *F. candida*, although there are small bacterial community abundance differences that can be detected (Fig. S4). To be specific, Proteobacteria was the first abundant and Firmicutes was the second abundant phylum in the yeast-derived F2 group, *F. graminearum*-derived F2 group Fg, and hunger-derived F2 group, respectively. In the F2 groups of *F. verticillioides*-derived, *A. nidulans*-derived, and *A. ochraceus*-derived *F. candida*, Firmicutes were the most abundant phylum and Proteobacteria was the second most abundant group. Moreover, Actinobacteria and Bacteroidetes were either third or fourth dominant groups (Table S4). Compared to F1 groups, the average abundance of Proteobacteria decreased to 43.47% from 66.01% but still was the most abundant phylum. In addition, the average abundance of Firmicutes represented the second dominant phylum, which increased to 36.19% from 4.23%. Followed by Actinobacteria (7.98%) and Bacteroidetes (7.10%). Notably, Actinobacteria abundance was significantly decreased as its content was 15.78% present as the second dominant phylum in F1 *F. candida*. Other bacterial taxa, such as Cyanobacteria, Chloroflexi and Fusobacteria, were only detected in minor proportions in all F2 groups.

  We further analyzed the sequencing result at the genus level (Fig. S5). Consistently, *Staphylococcus* (25.09%), *Stenotrophomonas* (5.34%), *Sphingobacterium* (3.11%), and *Pseudomonas* (3.07%) genera were annotated and the overall genus composition was similar among different groups (Fig. 6c). Specifically, *Staphylococcus* was the most dominant genus in yeast, *F. verticiiliodes*, *F. graminearum*, *A. nidulans,* and *A. ochraceus* groups, and detected as secondary dominant group in the hunger group. *Stenotrophomonas* was the most dominant genus in hunger and the second dominant genus in yeast. However, its abundance is less than 1.5% in the other groups. Similar to *Stenotrophomonas*, *Pseudomonas* was detected with higher abundance in yeast and hunger groups and the abundance was lower in *F. verticillioides*, *F. graminearum*, *A. nidulans,* and *A. ochraceus* groups. The secondary and third dominant bacterial genera were varied in mycotoxin-producing fungi feeding F2 groups, as *Escherichia* and *Microbacterium* were

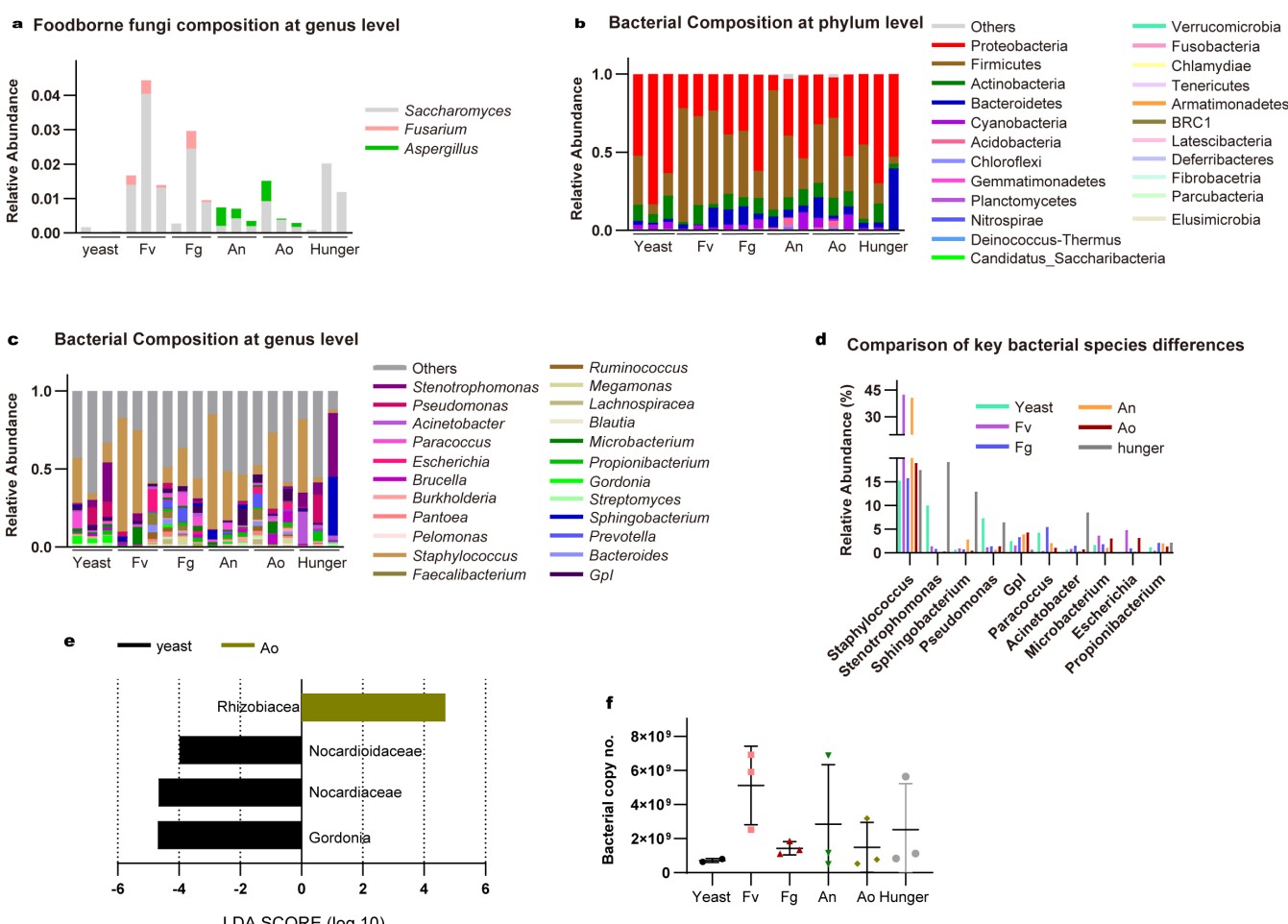

**FIG 6** The gut microbial community of F2 *F. candida* after each treatment was reintroduced with the yeast diet. The relative abundance of residue mycotoxin-producing fungi and foodborne yeast in the gut of F2 *F. candida* fed on yeast diets (a). The gut bacterial communities of F2 *F. candida* fed on yeast diets for 6 weeks originated from the different conditions of F1 *F. candida* that are classified at the phylum level (b) or genus level (c)  (d) Relative read abundance of the top 10 key bacteria in the F2 *F. candida* gut microbial community at the genus level. LEfSe analysis of microbial abundance between F2 *F. candida* groups (e) and bacterial content in the gut of 25–30 F2 *F. candida*. (f) reintroduced with the yeast diet. *P* values calculated using one-way ANOVA followed by Dunnett's multiple comparisons test.

detected in *F. verticillioides* group, *Paracoccus* and *Prevotella* were detected in *F. graminearum* group, GPI and *Sphingobacterium* were detected in *A. nidulans* group, and *Brucella* and GPI were detected in *A. ochraceus* group. *Sphingobacterium* was detected as the third dominant genera in the hunger group (Table S5). The other bacterial taxa were only detected in minor proportions in F2 groups.

LEfse method revealed that the family Rhizobiacea is significantly higher in the Ao group; the family Nocardioidaceae and Norcardiacear as well as its derivative genus *Gordonia* was enriched in the yeast group (Fig. 6e). According to these results, there are only one bacteria in the genus that differed and takes up a proportion of a rather low abundance (Fig. 6d). In addition, the absolute abundance of bacterial DNA amount was also restored when a normal yeast diet after fungal toxins exposure was provided (Fig. 6f). Overall, the bacterial community composition and dominant group in the F2 generation were restored to a composition similar to that of the yeast-fed groups. These results suggest that the diet of mycotoxin-producing fungi affects the microbiota of *F. candida*, but such effects can be recovered by the reintroduction of a yeast diet in the next generation at a very large level.

## DISCUSSION

In this study, we investigated the effects of grazing mycotoxin-producing fungi diets on *F. candida* growth parameters and gut microbiota across generations. We showed that grazing mycotoxin-producing fungi reduced springtail growth and disturbed gut microbial composition. Importantly, the growth and gut microbiota composition to a large extent could be restored when a normal diet was re-introduced. We also revealed that certain isolated gut bacteria are capable of inhibiting the growth of mycotoxin-producing fungi. Together, we found the growth and gut microorganisms are relatively dynamic and their changes may facilitate the animal host to tolerate mycotoxin-producing fungi.

Our findings were consistent with a previous report that showed that consuming mycotoxin-producing fungi as the only food source exhibits adverse effects on the body size of *F. candida*, and the survival rate is barely influenced (Fig. 1) (4, 11). In particular, when *F. candida* was fed on mycotoxin-producing fungi and hungered, their juvenile had a body length less than 0.5 mm, which is significantly shorter than those fed with yeast (1.5 mm), as *F. candida* of 1.5–3.0 mm in length is considered as maturity (32) (Fig. 1a through c). The reduction of juvenile body size under stress conditions is also reported (55). The juvenile number was largely increased in the *Fusarium* group than in the yeast group (Fig. 1e), similar results were reported by previous studies (14, 15). We also speculated that fixed-size living spaces (90 mm diameter Petri dishes) were provided and the juvenile size in the yeast group was bigger, thus leading to a limited number of offspring. In addition, the smallest size and limited juvenile number in *Aspergillus* groups imply that these fungi species have an inhibitory effect on the springtail population, which is probably due to the highly toxic metabolites produced by *Aspergillus* spp (4, 56). We also demonstrated that the negative effect of growth parameters caused by mycotoxin-producing fungi can be reversed in the F2 generation when a normal diet is re-supplied (Fig. 5). The re-supply of yeast to toxified and hunger collembola attenuated growth inhibition, as fitness parameters of F2 *F. candida* were restored. However, the F3 *F. candida* number was slightly lower in mycotoxin-producing fungi and hunger groups compared to the yeast group, which is probably due to the big disparity in size of the initial 30 F2 *F. candida* (Fig. 1c). These results indicate that *F. candida* populations present a strong tolerance for mycotoxin-producing fungi. However, Petri dish systems that provide fixed living conditions may not fully represent the field soil environment and different responses of soil collembolans to mycotoxin-producing fungi might be observed.

The gut microbiota of *F. candida* was profoundly changed when consuming mycotoxin-producing fungi, as gut bacterial 16S rRNA amplicon profiles at the phylum level were changed (Fig. 2a). Proteobacteria was predominant in the yeast and *Fusarium* groups. This bacteria phylum was reported as a dominant phylum in the microbiome of diverse soil invertebrates (35, 38, 55, 57–61). It can assist hosts in nutrient provisioning, detoxification, lignocellulose digestion, plastic degradation, and antimicrobial compound production (62, 63). Furthermore, *Fusarium* groups had an increased abundance of *Pseudomonas*, *Stenotrophomonas,* and *Brucella*, which largely overlapped with the yeast group. This implies that *Fusarium* groups harbor relatively balanced gut microbiota, which might contribute to the relatively better growth performance of these groups. By contrast, the *Aspergillus* and Hunger groups showed increased Actinobacteria and decreased Proteobacteria. At the genus level, *Aspergillus* groups had different bacteria genera *Microbacterium* and *Sphingopyxis* (Fig. 2e). *Microbacterium* were also enriched in the hunger group indicating their potential role in dealing with a deficiency of essential nutrients. *Microbacterium* was also the most abundant culturable bacteria genera across all the mycotoxin-producing fungal feeding groups (Fig. 3), its primary abundance is consistent with the previous finding (57). These findings suggest that *Aspergillus*-feeding groups are associated with a disrupted gut microbiota, which also correlates with poor growth performance. Our results further indicate a close link

between food quality, fitness parameters, and bacterial community composition (Table S7).

The composition of culturable gut microbiota in *F. candida* is also largely affected by eating mycotoxin-producing fungi, with Firmicutes dominating in the yeast group and Actinobacteria dominating in other groups. The Actinobacteria was predominantly isolated from *Aspergillus* and hunger groups, which is consistent with sequencing data of F1 *F. candida*. Actinobacteria play an important role in the development and maintenance of gut homeostasis and are involved in the modulation of gut permeability, immune system, and metabolism (64). In addition, Actinobacteria are widespread in diverse invertebrate species (57, 65) and act as defensive symbiosis in different insect taxa (66). Hence, the increased proportion of Actinobacteria observed in the mycotoxin-producing fungi and hunger groups suggests that Actinobacteria in those conditions may help the collembola to cope with those mycotoxin-producing fungi.

Springtails are relatively resilient to mycotoxin-producing fungi, as evidenced by the indeed ingest the mycotoxin-producing fungi when they served as the only diet, and the trace of these fungi can be mostly eliminated by the reintroduction of a yeast diet. Moreover, they can recover from fitness parameters and gut microbiota disruptions caused by mycotoxin-producing fungi diets. The administration of yeast to toxified collembola restored gut microbiota composition, as their gut microbiota composition and dominant microorganisms in F2 *F. candida* are similar to the yeast group (Fig. 6). In our study, a recovery of the gut microbiota of collembola was observed. It would be intriguing to investigate how animal hosts benefit from the microorganism community to cope with stressful conditions and achieve recovery under various stressful conditions. Such knowledge could potentially drive novel insights and innovative strategies for the therapy of various stressful stimuli. Our study provides a model to study the mechanism of an organism's gut microbiota recovery process. We also noticed that bacterial diversity in the F1 *F. candida* yeast group (Fig. 2a and b) differed from that in the F2 *F. candida* yeast group (Fig. 6a and b), as also noticed by previous study (57). This discrepancy may be explained by differences in culture environment, as we conducted our experiment at room temperature, which could have differed from the temperature and humidity in the laboratory.

In this study, we surprisingly found that some bacteria isolated from springtails fed on mycotoxin-producing fungi diets were able to inhibit the growth of mycotoxin-producing fungi. Specifically, co-culturing certain bacteria with *A. nidulans* and *A. ochraceus* resulted in a significantly lower fungi biomass (Fig. 4). Interestingly, the gut bacteria that inhibited mycotoxin-producing fungi growth in the *Aspergillus* group also correlated with poor fitness parameters and larger disruption in gut microbiota. By contrast, bacteria originating from *Fusarium* barely present an inhibition effect on mycotoxin-producing fungi growth. Therefore, we assumed that changes in animal physical parameters and gut microbiota positively correlated with the ability to inhibit mycotoxin-producing fungi growth. However, the gut environment and culture conditions are vastly different, and we are aware that only a small proportion of gut microbiota can be cultivated (Table S2). In our study, we were only able to isolate a small proportion of culturable microbes compared to the results from amplicon sequencing. In addition, our results are obtained from an experimental system, which might not reflect natural interactions. Nevertheless, the ability of gut bacteria to inhibit mycotoxin-producing fungi further highlighted that the altered gut microbes are important in alleviating the detrimental effect of eating mycotoxin-producing fungi.

Notably, springtail gut bacterial load is reduced by the consumption of mycotoxin-producing fungi and such reduction is recovered with the reintroduction of a normal diet (Fig. 2d through h and 6d through h). However, a stable core set of species present in the gut microbiota of *F. candida* regardless of the mycotoxin-producing fungi diet was detected. These bacteria species included *Brucella* (Brucellaceae), *Pseudomonas* (Pseudomonadaceae), *Methyloversatilis* (Sterolibacteriaceae), *Sphingopyxis* (Sphingomonadaceae), *Gordonia* (Gordoniacea), and *Microbacterium* (Microbacteriaceae) and have

been repeatedly detected at genus or family level in the gut microbiota of *F. candida* (35, 38, 55, 67).

In this study, we showed that springtails are relatively resilient to mycotoxin-producing fungi. Although mycotoxin-producing fungi diets had adverse effects on the growth of *F. candida* and their gut microbiota was significantly altered, the fitness parameter and gut microbiota of *F. candida* exhibited resilience and could recover from the adverse effects caused by the mycotoxin-producing fungi diet. Moreover, some of the culturable bacteria can inhibit fungi growth, indicating that alteration of gut microbiota could be a strategy employed by *F. candida* to defend against a mycotoxin-producing diet. These suggest that the changes in gut microbiota play an essential role in supporting the life of *F. candida*. Our study sheds light on the dynamic interactions between mycotoxin-producing fungi and the gut microbiota of springtails and has important implications for understanding the mechanisms underlying the regulation of gut bacterial composition in response to diverse stress conditions.

## ACKNOWLEDGMENTS

This work was supported by the National Natural Science Foundation of China (32001295 to Y.X., 31570388 to H.H.) and the Fundamental Research Funds for the Central Universities (the Starting Research Fund from the Northwest A&F University, No. 2452020037 to Y.X.). Y.X. is supported by Young Talent Fund of the University Association for Science and Technology in Shaanxi, China (No. 20210201).

Conceptualization, Y.X., H.H., L.X., and Data curation, Y.X., L.T., Z.X., and Formal analysis, Y.X., L.T., and Funding acquisition, Y.X., H.H., and Investigation, Y.X., L.T., Z.X., Y.H., X.D., K.W., J.Z., K.Y., and Methodology, Y.X., L.T., Project administration, H.H., and Resources Y.X., H.H., L.X., and Supervision, H.H., L.X., and Validation, Y.X., L.T., and Writing – original draft, Y.X., L.X., Writing – review & editing, H.H. All authors discussed the results and commented on the manuscript.

## AUTHOR AFFILIATIONS

[1]Key Laboratory of National Forestry and Grassland Administration for Control of Forest Biological Disasters in Western China, College of Forestry, Northwest A&F University, Yangling, Shaanxi, China
[2]College of Life Sciences, Northwest A&F University, Yangling, Shaanxi, China

## AUTHOR ORCIDs

Yang Xu  http://orcid.org/0000-0003-4252-0162
Lei Xu  http://orcid.org/0000-0002-9886-3677
Hong He  http://orcid.org/0000-0001-8267-3855

## FUNDING

| Funder | Grant(s) | Author(s) |
| --- | --- | --- |
| MOST | National Natural Science Foundation of China (NSFC) | 32001295 | Yang Xu |
| MOST | National Natural Science Foundation of China (NSFC) | 31570388 | Hong He |
| Starting Research Fund from the Northwest A&F University | 2452020037 | Yang Xu |
| Young Talent Fund of University Association for Science and Technology in Shaanxi, China | 20210201 | Yang Xu |

## AUTHOR CONTRIBUTIONS

Yang Xu, Conceptualization, Data curation, Formal analysis, Funding acquisition, Investigation, Methodology, Resources, Validation, Writing – original draft | Lingxiao Tang, Data curation, Formal analysis, Investigation, Methodology, Validation | Zhen Xie,

Data curation, Investigation | Xingwei Duan, Investigation | Kaisha Wang, Investigation | Jialin Zhu, Investigation | Yangyang Huang, Investigation | Kailang Yang, Investigation | Lei Xu, Conceptualization, Resources, Supervision, Writing – original draft | Hong He, Conceptualization, Funding acquisition, Project administration, Resources, Supervision, Writing – review and editing

## DATA AVAILABILITY

The sequences were deposited in GenBank under accession number as follows: F1 *F. candida* gut bacteria (PRJNA970098); F1 *F. candida* gut fungi (PRJNA970108); F2 *F. candida* gut bacteria (PRJNA970115); and F2 *F. candida* gut fungi (PRJNA970117). The sequences of culturable bacteria were deposited in GenBank under accession number: OQ933236-OQ933265.

## ADDITIONAL FILES

The following material is available online.

### Supplemental Material

**Supplemental material (Spectrum01035-24-S0001.docx).** Tables S1 to S7; Fig. S1 to S5.

### Open Peer Review

**PEER REVIEW HISTORY (review-history.pdf).** An accounting of the reviewer comments and feedback.

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
