## [Reviewer comments · Microbiology Spectrum]

Microbiology Spectrum

Effects of mycotoxin producing fungi on the fitness and gut bacterial community of the soil springtail *Folsomia candida*

Yang Xu, Lingxiao Tang, Zhen Xie, Xingwei Duan, Kaisha Wang, Jialin Zhu, Yangyang Huang, Kailang Yang, Lei Xu, and Hong He

Corresponding Author(s): Lei Xu, Northwest A&F University

Review Timeline:

Submission Date:	April 24, 2024
Editorial Decision:	May 31, 2024
Revision Received:	July 16, 2024
Editorial Decision:	August 23, 2024
Revision Received:	August 29, 2024
Accepted:	September 2, 2024

Editor: Patricia Albuquerque

Reviewer(s): Disclosure of reviewer identity is with reference to reviewer comments included in decision letter(s). The following individuals involved in review of your submission have agreed to reveal their identity: Lucio Navarro-Escalante (Reviewer #1)

Transaction Report:

DOI: <https://doi.org/10.1128/spectrum.01035-24>

Re: Spectrum01035-24 (Effects of toxic fungi on the fitness and gut bacterial community of the soil springtail *Folsomia candida*)

Dear Dr. Lei Xu:

Thank you for the privilege of reviewing your work. Below you will find my comments, instructions from the Spectrum editorial office, and the reviewer comments.

Revision Guidelines

Sincerely,
Patricia Albuquerque
Editor
Microbiology Spectrum

Reviewer #1 (Comments for the Author):

This research paper studies the effects of feeding on toxic fungi on the fitness and gut-associated microbiota of a soil springtail (*Folsomia candida*). Soil springtails were fed on several toxic fungal species (*Aspergillus*, *Fusarium*) and yeast (control treatment). The authors observed reduced insect fitness parameters and gut microbiota alterations when consuming toxic fungi. Additionally, switching back to a yeast diet restored fitness and microbiota community. Further assays using cultivated gut-

associated bacteria identified several bacterial isolates that inhibited toxic fungi growth under the experimental conditions. Therefore, authors claim that soil springtails "are relatively resilient to toxic fungi" and "some of the culturable bacteria can inhibit fungi growth, indicating that alteration of gut microbiota could be a strategy employed by *F. candida* to defend against a toxic diet".

However, there are several points that require more clarifications, including a number of statistical analyses for some experiments that need to be reported or need deeper attention in order to support several points or claims. The manuscript writing also requires attention for several typos, spelling errors, and some grammar issues.

Major points:

Reports of statistical test results are missing for several analyses and claims in the manuscripts. For example, comparisons of fitness parameters in figures 1 and 5, the comparison of absolute abundance of gut bacterial DNA amounts across feeding treatments (lines 344-345 and figure 2f; as well as line 472-474, figure 6f).

Classical Student's t-test analysis is not the most appropriate statistical test for analyzing insect preference in two-choice experiments. I suggest to apply other tests, such as nonparametric (e.g. Chi-square or Monte Carlo) to test hypotheses about insect preference in this type of assays.

It is a little odd that the numbers shown in the foodborne fungi composition analysis (<1%) (figure 2a and 6a) resulted in very low percentages for the target species (*Saccharomyces*, *Gibberella* and *Aspergillus*). What other fungal taxa with possibly higher abundances than the targets were detected in these analyses?

Figure descriptions need more details in order to better understand the data and graphs. These descriptions should be self-explanatory, so the reader can interpret the data without needing to read the main text.

The methodology indicates that fungal ITS primers (ITS1 and ITS4) were used for identification of culturable microorganisms (line 204), however in the results there is no mention about culturable fungi or any results from this fungal identification.

It is not clear what is the origin of the data presented in figure 4 for fungal DNA mass in the fungal inhibition assays. Are these from only DNA mass weighted as co-cultures (fungi/bacteria) or from qPCR absolute abundance?? Or a combination? Clarify this in the main text and the corresponding figures.

Minor points:

Keep consistent citation style in line 71.

Explain the motivation behind the two-choice experiments between rice and fungi

Check grammar for "nutrition" in line 78. Nutritional?

Check spelling for "hungried" (line 47) in description of supplementary figure S1

Check concordance of figure 2b in the text (line 330). Figure 2c?

Check x-axis labels in figure 2c: "Yeast" instead of "Others".

Spelling error for "enriching" (line 350).

Check grammar for line 352: "In terms number as well as" ?. Also in lines 258-359.

More detailed figures descriptions are needed for figures 4a and 4b.

Perhaps all percentages in lines 423 - 434 can be summarized in a table in the main manuscript or in supplementary.

Check spelling (line 521): "differed" or "different"?. Also in line 543: "albe"

Reviewer #2 (Comments for the Author):

The subject of the draft submitted by the authors is interesting. Interactions involving Collembola and fungi are relatively underexplored, especially when it comes to the resilience of collembolan species to the consumption of toxic fungi ». The draft

provides new information on these interactions and is important for the academic field.

1. Are the objectives of the study clearly stated?

Yes

2. Are the study methods (including theory/applicability/models) reported in sufficient detail to allow for their replicability and/or reproducibility?

Yes

3. Are statistical analyses, controls, sampling mechanism, and statistical reporting (e.g., P-values, CIs, effect sizes) appropriate and well described?

I feel like some test used are missing (The statistical analyses section is rather short compared to the number of experiments)

4. Is the clarity and the number of tables and figures in the manuscript appropriate?

Yes But a Table could be helpful to summarize some results.

5. Are the interpretation of results and study conclusions supported by the data and the study design?

Yes

6. In their discussion section, have the authors clearly emphasized the strengths of their study/theory/methods/argument?

Yes

7. In their discussion section, have the authors clearly emphasized the limitations of their study/theory/methods/argument?

They mention them, but it could be developed some more .

8. Is the manuscript structure (e.g., sections and its subheading), text flow, and writing appropriate?

Yes

9. Could the manuscript benefit from language editing?

English is not my native language, but I feel like the english is ok.

General comments :

Abstract :

Introduction :

If I understand correctly, the label « toxic fungi » refers to the toxicity for mammals (L74). *Aspergillus* is toxic for springtails, but is it the case for *Fusarium* species ? If so, make it clearer.

L68-71 add *H. nitidus* for Bourgeois et al. 2023. Moreover, Goncharov et al. 2020 is a meta analysis, so more springtail species are tested. This gives even more weight to the sentence as many species appear to consume these species.

L95-96 For 41, *Fusarium graminearum* is not really among the preferred species of *H. nitidus*, even though the *Collembola* can graze on it. For 42, it depends on the species so I would add some nuance to the sentence.

L103, remove « including » if you cite them all.

Material and methods

L119 Remove briefly

L121 Can you be more specific on the temperature ?

L122 Please indicate why springtails were starved

L134-135 please reformulate the sentence

L140 food, not diet

L158 reference needed for ImageJ software

L159 juveniles

L234 to suppress ?

L231 what test were used for the fitness parameters and F2 ?

Results

Some sections start or end with sentences that belong to the introduction or the discussion, as the Results section should only describe the results obtained.

Some results are quite dense (Gut bacterial communities of F2 *F. candida* after feeding with normal diets for example). Is there a way to present them as a table ?

Statistical analyses results should be written (test, p value) for every assertion.

L248-251 not needed in the results section

L254 Why are the results (and methods) of the food preference assay are in the supplementary informations ?

L266 suggesting that

L266 Interpretation should be in the discussion section

L351 significantly

L352 in terms of numbers

L406 « very lower abundance compared » You should indicate statistical test used to assert that one was lower than the other.

Discussion

The last three paragraphs are quite redundant and lack links to previous studies.

L499 You can link this assertion to the results of Rohlf et al., 2007 and Staden et al., 2011

L506 *F. candida* or *F. candida* populations ?

L507-508 Indeed, but can you elaborate why ?

L527-528 Can you link it to other studies demonstrating the same or similar results ?

L543 able

L548 I am not a specialist, but can you say that the animal is regulating its microbiota ?

The subject of the draft submitted by the authors is interesting. Interactions involving Collembola and fungi are relatively underexplored, especially when it comes to the resilience of collembolan species to the consumption of toxic fungi ». The draft provides new information on these interactions and is important for the academic field.

1. Are the objectives of the study clearly stated?

Yes

2. Are the study methods (including theory/applicability/models) reported in sufficient detail to allow for their replicability and/or reproducibility?

Yes

3. Are statistical analyses, controls, sampling mechanism, and statistical reporting (e.g., P-values, CIs, effect sizes) appropriate and well described?

I feel like some test used are missing (The statistical analyses section is rather short compared to the number of experiments)

4. Is the clarity and the number of tables and figures in the manuscript appropriate?

Yes But a Table could be helpful to summarize some results.

5. Are the interpretation of results and study conclusions supported by the data and the study design?

Yes

6. In their discussion section, have the authors clearly emphasized the strengths of their study/theory/methods/argument?

Yes

7. In their discussion section, have the authors clearly emphasized the limitations of their study/theory/methods/argument?

They mention them, but it could be developed some more .

8. Is the manuscript structure (e.g., sections and its subheading), text flow, and writing appropriate?

Yes

9. Could the manuscript benefit from language editing?

English is not my native language, but I feel like the english is ok.

General comments :

Abstract :

Introduction :

If I understand correctly, the label « toxic fungi » refers to the toxicity for mammals (L74). *Aspergillus* is toxic for springtails, but is it the case for *Fusarium* species ? If so, make it clearer.

L68-71 add *H. nitidus* for Bourgeois *et al.* 2023. Moreover, Goncharov *et al.* 2020 is a meta analysis, so more springtail species are tested. This gives even more weight to the sentence as many species appear to consume these species.

L95-96 For 41, *Fusarium graminearum* is not really among the preferred species of *H. nitidus*, even though the Collembola can graze on it. For 42, it depends on the species so I would add some nuance to the sentence.

L103, remove « including » if you cite them all.

Material and methods

L119 Remove briefly

L121 Can you be more specific on the temperature ?

L122 Please indicate why springtails were starved

L134-135 please reformulate the sentence

L140 food, not diet

L158 reference needed for ImageJ software

L159 juveniles

L234 to suppress ?

L231 what test were used for the fitness parameters and F2 ?

Results

Some sections start or end with sentences that belong to the introduction or the discussion, as the Results section should only describe the results obtained.

Some results are quite dense (Gut bacterial communities of F2 *F. candida* after feeding with normal diets for example). Is there a way to present them as a table ?

Statistical analyses results should be written (test, *p* value) for every assertion.

L248-251 not needed in the results section

L254 Why are the results (and methods) of the food preference assay are in the supplementary informations ?

L266 suggesting that

L266 Interpretation should be in the discussion section

L351 significantly

L352 in terms of numbers

L406 « very lower abundance compared » You should indicate statistical test used to assert that one was lower than the other.

Discussion

The last three paragraphs are quite redundant and lack links to previous studies.

L499 You can link this assertion to the results of Rohlf's et al., 2007 and Staaden et al., 2011

L506 *F. candida* or *F. candida* populations ?

L507-508 Indeed, but can you elaborate why ?

L527-528 Can you link to other studies demonstrating the same or similar results ?

L543 able

L548 I am not a specialist, but can you say that the animal is regulating its microbiota ?

RESPONSE TO THE REVIEWERS' COMMENT ON THE MANUSCRIPT

"Effects of toxic fungi on the fitness and gut bacterial community of the soil springtail *Folsomia candida*"

Referee: plain black text

Authors: plain blue text

We would like to express our gratitude to all three reviewers for their valuable and constructive feedback. Below, we provide detailed responses to each of their comments.

Reviewer #1 (Remarks to the Author):

Comment:

This research paper studies the effects of feeding on toxic fungi on the fitness and gut-associated microbiota of a soil springtail (*Folsomia candida*). Soil springtails were fed on several toxic fungal species (*Aspergillus*, *Fusarium*) and yeast (control treatment). The authors observed reduced insect fitness parameters and gut microbiota alterations when consuming toxic fungi. Additionally, switching back to a yeast diet restored fitness and microbiota community. Further assays using cultivated gut-associated bacteria identified several bacterial isolates that inhibited toxic fungi growth under the experimental conditions. Therefore, authors claim that soil springtails "are relatively resilient to toxic fungi" and "some of the culturable bacteria can inhibit fungi growth, indicating that alteration of gut microbiota could be a strategy employed by *F. candida* to defend against a toxic diet".

However, there are several points that require more clarifications, including a number of statistical analyses for some experiments that need to be reported or need deeper attention in order to support several points or claims. The manuscript writing also requires attention for several typos, spelling errors, and some grammar issues.

Response:

Thank you for your valuable feedback on our manuscript. We appreciate your careful review and the constructive comments provided. We thank you for your positive comment that the effects of feeding on toxic fungi on the fitness and gut-associated microbiota of the soil springtail (*Folsomia candida*) are of interest. We agree that more detailed statistical analyses are necessary to support our claims. In the revised manuscript, we have included a comprehensive description of the statistical methods used for each experiment. Also, additional statistical tests were performed to ensure the robustness of the results. We have thoroughly revised the manuscript to address typographical errors, spelling mistakes, and

grammatical issues. We greatly appreciate your insightful comments and the opportunity to improve our manuscript. We hope you will find our revised manuscript satisfactory.

Major points:

Comment 1:

Reports of statistical test results are missing for several analyses and claims in the manuscripts. For example, comparisons of fitness parameters in figures 1 and 5, the comparison of absolute abundance of gut bacterial DNA amounts across feeding treatments (lines 344-345 and figure 2f; as well as line 472-474, figure 6f).

Response:

Thank you for your comments. For Fig. 1 and Fig. 5, the One-way ANOVA followed by Dunnett's multiple comparisons was used to compare fitness parameters. The comparison of the absolute abundance of gut bacterial DNA amounts across feeding treatments in Fig. 2f and Fig. 6f was also performed by using One-way ANOVA with Dunnett's multiple comparisons. However, there is no significant difference between the absolute abundance of gut bacterial DNA amounts across feeding treatments (Fig. 2f and Fig. 6f). In addition, we have also clearly stated the statistical methods used for each comparison in the material and method section and figure legends. Thank you again for your valuable feedback.

Comment 2:

Classical Student's t-test analysis is not the most appropriate statistical test for analyzing insect preference in two-choice experiments. I suggest to apply other tests, such as nonparametric (e.g. Chi-square or Monte Carlo) to test hypotheses about insect preference in this type of assays.

Response:

Thank you for your comments. The Mann-Whitney test was employed to establish the variation in springtail preference between each toxic fungi and rice medium. This statistical test was also used in other studies to test food preferences (Li et al., 2022; Pangemanan et al., 2023).

Relevant References:

Li M, Tan HE, Lu Z, Tsang KS, Chung AJ, Zuker CS. Gut-brain circuits for fat preference. *Nature*. 2022;610(7933):722-730. doi:10.1038/s41586-022-05266-z
Pangemanan L, Irwanto I, Maramis MM. Psychological dominant stressor modification to an animal model of depression with chronic unpredictable mild stress. *Vet World*. 2023;16(3):595-600. doi:10.14202/vetworld.2023.595-600

Comment 3:

It is a little odd that the numbers shown in the foodborne fungi composition analysis (<1%) (figure 2a and 6a) resulted in very low percentages for the target species (*Saccharomyces*,

Gibberella and Aspergillus). What other fungal taxa with possibly higher abundances than the targets were detected in these analyses?

Response:

Thank you for pointing out this important issue. We understand your concern regarding very low percentages for the target species like *Saccharomyces*, *Gibberella* and *Aspergillus* in Fig. 2a and Fig. 6a. To explain this, we have shown the sequencing results below (R#Figure 1). In F1 *F. candida* gut fungi community, the relative abundance of yeast (0.18%), *F. graminearum* (0.81%) are very low, whereas the abundance of *F. verticillioides* (4.87%), *A. nidulans* (10.09%) and *A. ochraceus* (69.22%) is higher, which might due to *F. candida* have different digestion capabilities for yeast and different toxic fungi. In F2 *F. candida*, the relative abundance of all foodborne fungi is very low. As shown in R#Figure 1, there is a very high abundance of “Others” (unclassified fungi sequence) in the fungi sequence data, and also very few other fungi species can be detected. However, this large proportion of “Others” led to the panel being hard to read and may distract the reader’s attention. Thus, we only present the foodborne fungi abundance to support that the fungi were uptake by *F. candida*.

R#Figure 1. The gut fungi communities of F1 *F. candida* feed on yeast, toxic fungi, and hungered are classified at the phylum level (a) and genus level (b). The fungi bacterial communities of F2 *F. candida* fed on yeast diets for six weeks originated from the different conditions of F1 *F. candida* that are classified at the phylum level (c) or genus level (d).

Comment 4:

Figure descriptions need more details in order to better understand the data and graphs. These descriptions should be self-explanatory, so the reader can interpret the data without needing to read the main text.

Response:

We appreciate your suggestion. In response to your comment, we have revised the figure descriptions in our manuscript to include more details that will aid in better understanding the data and graphs. Please see Line 778 to 825 in the new version.

Comment 5:

The methodology indicates that fungal ITS primers (ITS1 and ITS4) were used for identification of culturable microorganisms (line 204), however in the results there is no mention about culturable fungi or any results from this fungal identification.

Response:

We apologize for this mistake. We have removed this.

Comment 6:

It is not clear what is the origin of the data presented in figure 4 for fungal DNA mass in the fungal inhibition assays. Are these from only DNA mass weighted as co-cultures (fungi/bacteria) or from qPCR absolute abundance?? Or a combination? Clarify this in the main text and the corresponding figures.

Response:

Thank you for your comments. The data presented in Figure 4 for fungal DNA mass in the fungal inhibition assays are based on the combination of qPCR absolute abundance and mass weight (fungi and bacteria). For this result, we collected the dual culture of toxic fungi and individual culturable microbe, then freeze-dried the samples, and the total weight was determined. After that, 10 mg of freeze-dried samples were weighted for DNA extraction. Then, the standard made by each toxic fungi was used to determine the toxic fungi DNA copy number in 10 mg dried material. Thirdly, we multiply the copy number of 10 mg with total weight/10 mg to get the total copy number of toxic fungi. We have revised the main text and corresponding figures to explicitly clarify that the data in Figure 4 represents qPCR absolute abundance of fungal DNA mass in the fungal inhibition assays.

Minor points:

Comment 1:

Keep consistent citation style in line 71.

Response:

We feel sorry for this mistake, this has been modified as suggested.

Comment 2:

Explain the motivation behind the two-choice experiments between rice and fungi

Response:

The rationale for performing the two-choice experiments between rice and fungi is to remove the influence of the rice substrate medium. In this study, the fungi diet was prepared by inoculating the fungi on a very thin layer of rice medium and cultured at 25°C for 7 days. The fungi occupied the rice medium completely in 7 days. However, to exclude the possibility that springtails might feed on rice medium instead of toxic fungi, we performed this experiment to explain springtails will prefer a fungi diet rather than rice medium.

Comment 3:

Check grammar for "nutrition" in line 78. Nutritional?

Response:

Modified as suggested.

Comment 4:

Check spelling for "hungried" (line 47) in description of supplementary figure S1

Response:

We feel sorry for this mistake, this has been changed to hungry.

Comment 5:

Check concordance of figure 2b in the text (line 330). Figure 2c?

Response:

We feel sorry for this mistake, this has been modified as suggested.

Comment 6:

Check x-axis labels in figure 2c: "Yeast" instead of "Others".

Response:

Modified as suggested.

Comment 7:

Spelling error for "enriching" (line 350).

Response:

We feel sorry for this typo, this has been corrected.

Comment 8:

Check grammar for line 352: "In terms number as well as"?. Also in lines 258-359.

Response:

We feel sorry for these mistakes, this has been modified as suggested.

Comment 9:

More detailed figures descriptions are needed for figures 4a and 4b.

Response:

Thank you for your suggestions. We have added more figure descriptions for Figure 4a and 4b.

Comment 10:

Perhaps all percentages in lines 423 - 434 can be summarized in a table in the main manuscript or in supplementary.

Response:

Thank you for your suggestions. We have summarized the data in Supplementary Table 4 in the new version.

Comment 11:

Check spelling (line 521): "differed" or "different"?. Also in line 543: "albe"

Response:

We apologize for these typos. We have changed "differed" to "different" and "albe" to "able".

Reviewer #2 (Comments for the Author):

The subject of the draft submitted by the authors is interesting. Interactions involving Collembola and fungi are relatively underexplored, especially when it comes to the resilience of collembolan species to the consumption of toxic fungi ». The draft provides new information on these interactions and is important for the academic field.

1. Are the objectives of the study clearly stated?

Yes

2. Are the study methods (including theory/applicability/models) reported in sufficient detail to allow for their replicability and/or reproducibility?

Yes

3. Are statistical analyses, controls, sampling mechanism, and statistical reporting (e.g., P-values, CIs, effect sizes) appropriate and well described?

I feel like some test used are missing (The statistical analyses section is rather short compared to the number of experiments)

4. Is the clarity and the number of tables and figures in the manuscript appropriate?

Yes But a Table could be helpful to summarize some results.

5. Are the interpretation of results and study conclusions supported by the data and the study design?

Yes

6. In their discussion section, have the authors clearly emphasized the strengths of their study/theory/methods/argument?

Yes

7. In their discussion section, have the authors clearly emphasized the limitations of their study/theory/methods/argument?

They mention them, but it could be developed some more.

8. Is the manuscript structure (e.g., sections and its subheading), text flow, and writing appropriate?

Yes

9. Could the manuscript benefit from language editing?

English is not my native language, but I feel like the English is ok.

Response:

Thank you for your valuable feedback and suggestions on our manuscript. We appreciate your time and effort in reviewing our work.

General comments:

Abstract :

Comment 1:

Introduction :

If I understand correctly, the label « toxic fungi » refers to the toxicity for mammals (L74). Aspergillus is toxic for springtails, but is it the case for Fusarium species? If so, make it clearer.

Response:

Thank you for your valuable feedback on our manuscript. We appreciate your clarification on the term toxic fungi and the distinction between toxicity for mammals and other organisms such as springtails. We have rephrased the description accordingly.

Comment 2:

L68-71 add H. nitidus for Bourgeois et al. 2023. Moreover, Goncharov et al. 2020 is a meta analysis, so more springtail species are tested. This gives even more weight to the sentence as many species appear to consume these species.

Response:

Thank you for your suggestions. We have added these important references to our manuscript.

Comment 3:

L95-96 For 41, *Fusarium graminearum* is not really among the preferred species of *H. nitidus*, even though the Collembola can graze on it. For 42, it depends on the species so I would add some nuance to the sentence.

Response:

Thank you for your valuable suggestions. We have rephrased the description accordingly.

Comment 4:

L103, remove « including » if you cite them all.

Response:

We have removed “including”.

Comment 5:

Material and methods

L119 Remove briefly

Response:

Removed as suggested.

Comment 6:

L121 Can you be more specific on the temperature?

Response:

The temperature is 15-25°C. We have added this information.

Comment 7:

L122 Please indicate why springtails were starved

Response:

Springtails were starved to increase their appetite. We have added this information in the manuscript.

Comment 8:

L134-135 please reformulate the sentence

Response:

Thank you for your suggestions. We have rephrased the description accordingly.

Comment 9:

L140 food, not diet

Response:

Modified as suggested.

Comment 10:

L158 reference needed for ImageJ software

Response:

We have added this information in the new version.

Comment 11:

L159 juveniles

Response:

Modified as suggested.

Comment 12:

L234 to suppress?

Response:

We have changed “suppress” to inhibit”.

Comment 13:

L231 what test were used for the fitness parameters and F2?

Response:

The fitness parameters in F1 and F2 were analyzed by using the One-way ANOVA test. We have added this information.

Comment 14:

Results

Some sections start or end with sentences that belong to the introduction or the discussion, as the Results section should only describe the results obtained.

Some results are quite dense (Gut bacterial communities of F2 *F. candida* after feeding with normal diets for example). Is there a way to present them as a table?

Response:

Thank you for your suggestions and we appreciate your comments. We have carefully reviewed and revised these sections to ensure they are appropriately structured. Regarding the density of some results, such as the description of gut bacterial communities of F2 *F. candida* after feeding with normal diets, we have presented this information in a table format would enhance clarity and readability. Please see the new Supplementary Table 4 and Supplementary Table 5.

Comment 15:

Statistical analyses results should be written (test, p value) for every assertion.

Response:

Thank you for your comments. We have used a lot of statistical analysis in this manuscript and it may be a distraction to write the p-value for every assertion in the results section. We have shown the statistical analyses we used and the use of an asterisk for each statistical analysis (* $P < 0.05$; ** $P < 0.01$; *** $P < 0.001$) in each figure legend.

Comment 16:

L248-251 not needed in the results section

Response:

Thank you for your suggestions. We have removed this.

Comment 17:

L254 Why are the results (and methods) of the food preference assay are in the supplementary informations?

Response:

Thank you for your comment. The food preference assay is aimed to exclude the effect of rice medium on springtails. The results proved that springtails will feed on toxic fungi than rice medium. This result is not directly related to springtails, toxic fungi, or gut bacterial community, therefore we put the results (and methods) of the food preference assay in the Supplementary Information.

Comment 18:

L266 suggesting that

Response:

Thank you for your suggestions. We have changed this.

Comment 19:

L266 Interpretation should be in the discussion section

Response:

Thank you for your suggestions. We have moved this to the discussion section.

Comment 20:

L351 significantly

Response:

Modified as suggested.

Comment 21:

L352 in terms of numbers

Response:

Thank you for your suggestions. We have changed “number” to “numbers”.

Comment 22:

L406 « very lower abundance compared » You should indicate statistical test used to assert that one was lower than the other.

Response:

Thank you for your suggestions. We have performed statistical tests to support this claim. Please see the new Supplementary Table 6.

Comment 23:

Discussion

The last three paragraphs are quite redundant and lack links to previous studies.

Response:

Thank you for your comments. For the paragraphs from Line 539 to 553, we are aiming to elucidate the role of gut microbiota in assisting the host in detoxifying toxic fungi, which could pave the way for discovering more functional microbes within the invertebrate gut microbiota. Given the lack of closely related studies, we are pioneering this investigation and setting a foundation for future research in this area. For the paragraphs from Line 555 to 562, we are trying to discuss that identifying the core gut microorganism species in *F. candida* provides crucial foundational information for future studies. This knowledge will enable further investigation into the gut microorganism community composition, diversity, and function in springtails. Our findings align with previous studies, enhancing our understanding of these microbial communities. The last section provides a summary of our article and we believe it may help the reader to summarize the study.

L499 You can link this assertion to the results of Rohlf et al., 2007 and Staaden et al., 2011

Response:

Thank you for your suggestions. We have added these references.

Comment 24:

L506 *F. candida* or *F. candida* populations?

Response:

We have changed “*F. candida*” to “*F. candida* populations”.

Comment 25:

L507-508 Indeed, but can you elaborate why?

Response:

Thank you for pointing out this. We speculated that several reasons can explain this. First, there is only limited space are provided and this may suppress the reproductivity of *F. candida*. Second, other organisms like animals, invertebrates, bacteria, and fungi may contribute to interaction in soil ecosystems. Thus, we think Petri dish systems may not fully recapitulate the field soil environment.

Comment 26:

L527-528 Can you link it to other studies demonstrating the same or similar results?

Response:

Thank you for your comments. We have searched the literature and did not find other studies demonstrating similar results. To support this claim, we employed Spearman analysis to indicate the correlation between diet type, fitness parameters and gut microbiota changes. As shown in the new Supplementary Table 7, there is a significant correlation between diet type, F1 *F. candida* length, F1 *F. candida* width, number of different bacteria (from LEfSe analysis) and different bacterial abundance.

Comment 27:

L543 able

Response:

Thank you. We have corrected this.

Comment 28:

L548 I am not a specialist, but can you say that the animal is regulating its microbiota?

Response:

Thank you for your comment. We have changed “regulating” to “benefit from”.

Re: Spectrum01035-24R1 (Effects of toxic fungi on the fitness and gut bacterial community of the soil springtail *Folsomia candida*)

Dear Dr. Lei Xu:

Thank you for the privilege of reviewing your work. Below you will find my comments, instructions from the Spectrum editorial office, and the reviewer comments.

I only suggest that you replace "toxic fungi" for "Mycotoxin producing fungi" in the manuscript title and throughout the manuscript text. I believe that would increase clarity.

Line 32 - Please specify/clarify what physiological parameters were decreased.

Line 404 - *F. graminearum* and *A. nidulans* was times lower in F2 *F. candida*

How many times lower?

Revision Guidelines

Sincerely,
Patricia Albuquerque
Editor
Microbiology Spectrum

Reviewer #2 (Comments for the Author):

The authors addressed my comments on the first draft to send an improved version. I read it with great interest and I think they bring some important data to the growing academic field of -Collembola-Fungi interactions.

Minor comments :

Abstract :

Introduction :

L61 Mold is a type a structure produced by some fungi, not a group of fungi.

L72 effect

Material and methods

L161 the article to cite for ImageJ: Schneider, C. A., Rasband, W. S., & Eliceiri, K. W. (2012). NIH Image to ImageJ: 25 years of image analysis. *Nature methods*, 9(7), 671-675.

L236 what are "all other experiments"?

L260 I think a verb is missing for the sentence to be correct

L269 increased seems more appropriate here

Results

L293-296 In my opinion these sentences do not belong in the result section

Figures 1a and 2b: I am not a huge fan of cutting the Y axis as it can be hard to appreciate the differences. Can you find a solution for that?

Discussion

L460 We showed that a toxic fungi diet/ We showed that grazing toxic fungi

L467 that showed that consuming

L479 what is an aggressive toxin?

L482 The sentence does not make sense to me, are you sure it is correct?

L504 dealing with hunger or a deficiency for some nutrients?

L527 are similar to what?

RESPONSE TO THE REVIEWERS' COMMENT ON THE MANUSCRIPT

"Effects of toxic fungi on the fitness and gut bacterial community of the soil springtail *Folsomia candida*"

Referee: plain black text

Authors: plain blue text

We would like to express our gratitude to the editor and reviewers for their valuable and constructive feedback. Below, we provide detailed responses to each of their comments. As suggested, we have modified the title to "Effects of mycotoxin producing fungi on the fitness and gut bacterial community of the soil springtail *Folsomia candida*".

Comment from editor

Comment 1:

I only suggest that you replace "toxic fungi" for "Mycotoxin producing fungi" in the manuscript title and throughout the manuscript text. I believe that would increase clarity.

Response:

Thank you for your suggestion. We have replaced the "toxic fungi" with "Mycotoxin producing fungi" in the title and throughout the manuscript text.

Comment 2:

Line 32 - Please specify/clarify what physiological parameters were decreased.

Response:

Thank you for your comment. The physiological parameters include body size and reproductive ability. We have added this in the new version of the Abstract.

Comment 3:

Line 404 - *F. graminearum* and *A. nidulans* was times lower in F2 *F. candida*

How many times lower?

Response:

We apologize for the typo of this sentence. It is "4-31 times". We have added this in the new version.

Reviewer #2 (Comments for the Author):

The authors addressed my comments on the first draft to send an improved version. I read it with great interest and I think they bring some important data to the growing academic field of - Collembola-Fungi interactions.

Response:

Thank you very much for your thoughtful feedback and for taking the time to review our revised manuscript. We greatly appreciate your positive comments regarding our work on Collembola and fungi interactions and are delighted to hear that you found our data to be important to this growing academic field. We are committed to making further improvements to the manuscript and will ensure that all relevant modifications are included in the final version.

Minor comments:

Abstract :

Comment 1:

Introduction :

L61 Mold is a type a structure produced by some fungi, not a group of fungi.

Response:

Thank you for your suggestions. We have rephrased this description accordingly.

Comment 2:

L72 effect

Response:

We have removed "effect".

Comment 3:

Material and methods

L161 the article to cite for ImageJ: Schneider, C. A., Rasband, W. S., & Eliceiri, K. W. (2012). NIH Image to ImageJ: 25 years of image analysis. Nature methods, 9(7), 671-675.

Response:

Thank you for your suggestions. We have added this reference.

Comment 4:

L236 what are "all other experiments"?

Response:

Thank you for your suggestions. We have corrected this.

Comment 5:

L260 I think a verb is missing for the sentence to be correct

Response:

Thank you for your suggestions. We have corrected this.

Comment 6:

L269 increased seems more appropriate here

Response:

Thank you for your suggestions. We apologize for the typo and have modified it as suggested.

Comment 7:

Results

L293-296 In my opinion these sentences do not belong in the result section

Response:

We apologize for the confusion. We have removed some of the sentences.

Comment 8:

Figures 1a and 2b: I am not a huge fan of cutting the Y axis as it can be hard to appreciate the differences. Can you find a solution for that?

Response:

Thank you for your suggestions. If we understand correctly, we assume you are referring to Figure 1b and Figure 2a, in which we used two segments for the Y-axis. To address this, we used the figure with the full Y-axis. Please see the new version of Figure 1 and 2.

Comment 9:

Discussion

L460 We showed that a toxic fungi diet/ We showed that grazing toxic fungi

Response:

Modified as suggested.

Comment 10:

L467 that showed that consuming

Response:

Modified as suggested.

Comment 11:

L479 what is an aggressive toxin?

Response:

We changed “aggressive toxin” to “highly toxic metabolites”.

Comment 12:

L482 The sentence does not make sense to me, are you sure it is correct?

Response:

We apologize for the confusion. We have replaced “administration” with “re-supply”.

Comment 13:

L504 dealing with hunger or a deficiency for some nutrients?

Response:

Thank you for your suggestions. We have modified this as suggested.

Comment 14:

L527 are similar to what?

Response:

Thank you for your comment. In this sentence, we aimed to say that dominant microorganisms in F2 *F. candida* are similar to the yeast group. We have added this.

Re: Spectrum01035-24R2 (Effects of mycotoxin producing fungi on the fitness and gut bacterial community of the soil springtail *Folsomia candida*)

Dear Dr. Lei Xu:

Your manuscript has been accepted, and I am forwarding it to the ASM production staff for publication. Your paper will first be checked to make sure all elements meet the technical requirements. ASM staff will contact you if anything needs to be revised before copyediting and production can begin. Otherwise, you will be notified when your proofs are ready to be viewed.

Sincerely,
Patricia Albuquerque
Editor
Microbiology Spectrum